# Relationship between the asymmetry of the resting scapular position and the prevalence of latent myofascial trigger points in the trapezius muscle in asymptomatic adults

**Yasuhiro Aki[1,2], Ippei Okino[2], Genki Tsuyama[2], Yoshiyasu Tanitsu[2], Hisao Nishijo[1], Kouichi Takamoto**[1]*

**1** Department of Sports and Health Sciences, Faculty of Human Sciences, University of East Asia, Yamaguchi, Japan, **2** Givers Corporation, Tokyo, Japan

\* ktakamo@toua-u.ac.jp

## Abstract

Myofascial trigger points (MTrPs) and body postural misalignment, including the position of the scapula, can contribute to the onset and persistence of musculoskeletal pain. However, the relationship in asymptomatic cases remains unclear. Therefore, this study aimed to investigate the relationship between the asymmetry of the resting scapular position and latent MTrPs in the upper trapezius muscle (UTM) in asymptomatic adults. A total of 32 asymptomatic adult men (mean age, 26.28 ± 1.1 years) were included in this study. Full-body photographs were taken from the posterior view, with the participants resting in a standing position. To determine the degree of asymmetry of the resting scapular position, the horizontal scapular alignment angle (HSAA) was analyzed from the photographs. The assessor identified the presence of latent MTrPs in the right and left UTMs. The HSAA was significantly lower in the group with latent MTrPs in the right UTM than in those without latent MTrPs. The results showed that the right scapula was more depressed than the left scapula in the group with latent MTrPs in the right UTM. Furthermore, multiple regression analysis indicated that the dominant arm and presence of latent MTrPs in the right UTM significantly contributed to the prediction of the HSAA. The results of this study demonstrated a close relationship between the asymmetry of the resting scapular position and latent MTrPs in the UTM in asymptomatic adults, which may contribute to the onset and persistence of musculoskeletal pain.

## Introduction

Myofascial trigger points (MTrPs) are sensitive points in the muscle taut band and are considered one of the factors associated with the onset and chronicity of musculoskeletal pain [1–3]. MTrPs are classified into active and latent MTrPs [1]. Active

**Data availability statement:** Datasets are available in the Zenodo database (DOI: https://doi.org/10.5281/zenodo.17155991).

**Funding:** This work was supported by Givers Corporation (https://givers.co.jp/) and a Japan Society for the Promotion of Science (JSPS) Grant-in-Aid for Scientific Research (C) (23K10467) (https://www.jsps.go.jp/j-grant-sinaid/) awarded to K.T. The funder, Givers Corporation, was involved in the study design, data collection, analysis, interpretation of data, writing of the manuscript, and the decision to submit the paper for publication. The JSPS grant had no role in study design, data collection and analysis, decision to publish, or preparation of the manuscript.

**Competing interests:** The authors have declared that no competing interests exist.

MTrPs induce local or remote pain spontaneously or by compression, which is consistent with the patient's complaint. Conversely, latent MTrPs induce local or remote pain by compression, which is not consistent with the patient's complaint. Active MTrPs are observed only in patients with musculoskeletal pain, such as pain in the lower back, neck, and knees, whereas latent MTrPs are observed not only in patients with musculoskeletal pain but also in asymptomatic individuals [4–6]. The mechanism of MTrP formation has been proposed to involve mechanical stress from repetitive or sustained movements, which overload the muscle and result in sustained, localized muscle contractions (muscle taut bands) [1–3]. These sustained muscle contractions cause an energy crisis due to local ischemia and increased metabolic energy demands, causing the release of painful substances and formation of hypersensitivity points in the muscle taut band (latent MTrPs). The persistence of the above process results in the transition to active MTrPs and induces pain. Furthermore, it leads to the permanence of MTrPs and pain chronicity. However, the mechanical stress that results in the permanence of MTrPs and pain chronicity is not well identified.

Body postural misalignment is associated with the development and permanence of MTrPs [3,7,8]. Body postural misalignment is characterized by the disruption of the normal relative positional relationships among body parts, such as the spine, scapula, and pelvis, which can be present in both symptomatic and asymptomatic cases [9]. This postural misalignment is also attributed to the dysfunction of muscles related to posture control caused by the change in muscle length resulting from repetitive and sustained movements with an unnatural posture [9–12]. Poor posture has been suggested to further cause muscle weakness and imbalance, which adds mechanical stress to the musculoskeletal system, inducing musculoskeletal pain [10,13–18]. Thus, mechanical stress to muscles induced by the postural misalignment may be involved in MTrP development and permanence. However, muscle dysfunctions, such as muscle weakness and imbalance, changes in muscle recruitment patterns, and limited range of motion, were observed in both latent and active MTrPs [3,19–22]. Thus, MTrPs may lead to body postural misalignment. These findings indicate a close relationship between MTrPs and body postural misalignment, which is involved in the onset and persistence of musculoskeletal pain.

Previous studies have reported relationships between postural alignment and MTrPs in patients with chronic musculoskeletal pain [23–26]. However, these relationships in asymptomatic cases remain unclear. The asymmetry of the resting scapular position is an assessment of the postural alignment. The position of the scapula is suggested to be influenced by the dysfunction of scapulothoracic muscles, such as the trapezius, rhomboid, and serratus anterior muscles, and bone, joint, and neurologic problems, such as thoracic kyphosis and long thoracic nerve palsy [27,28]. In particular, the trapezius muscle is involved in the stabilization of the scapula, and weakness in the upper region of the trapezius muscle was suggested to affect the depression of the scapular position [29]. In asymptomatic individuals without musculoskeletal pain, a difference was observed in the heights of the right and left scapulae, indicating that one scapula was depressed [30]. The latent MTrPs in the upper trapezius muscle (UTM) are common in asymptomatic people [5]. A depressed

scapular position was suggested to result in an extended UTM position, which leads to increased muscle tension [9]. Because sustained muscle contractions involve the formation of MTrPs [1–3], the increased muscle tension caused by the depressed scapula may affect MTrP formation. Conversely, MTrPs induce muscle weakness, which may change the position of the scapula [3]. Thus, the asymmetry of the resting scapular position in asymptomatic cases may be associated with the presence of MTrPs in the UTM. Therefore, this study aimed to investigate the relationship between the resting scapular position and the presence of latent MTrPs in asymptomatic men.

## Methods

### Participants

A total of 32 asymptomatic adult men (mean age, 26.28 ± 1.1 years) were included in this study. The participants were recruited between March 2023 and August 2023. Adult men without musculoskeletal disorders, pain originating from those disorders, and medical illness were included, whereas those with a history of musculoskeletal injury and internal diseases within 3 months were excluded from the study. No specific exclusion criteria were applied based on physical activity levels or dominant-side sport-specific training.

This study complies with the Helsinki Declaration. The experimental protocols were reviewed and approved by the Ethics Assessment Committee for Research Involving Human Subjects at the University of East Asia (approval no. 2022−3; approved on April 15, 2022). Written informed consent was obtained from all participants.

### Study procedures

A full-body photograph of each participant was taken from the posterior view, with the participant resting in a standing position. The assessor identified the presence and number of latent MTrPs in the left and right UTMs. The degree of asymmetry of the resting scapular position was analyzed using photograph analysis software.

### Assessment of the asymmetry of the resting scapular position

The participants were instructed to maintain a standing posture, and a sticker was placed on the right and left medial spine of the scapula (MSS) as a landmark. Full-body digital photographs were taken from the posterior view using an iPhone 12 Pro camera, with the participants resting in a standing position. Image data were imported from the iPhone to a computer, and photographic analysis was performed using Kinovea (https://www.kinovea.org/). To determine the degree of asymmetry of the resting scapular position, the horizontal scapular alignment angle (HSAA) was analyzed from the photograph. The HSAA was defined as the angle between the horizontal line from the left MSS and the straight line connecting the left and right MSS (Fig 1).

### MTrP assessment

A single assessor with 10 years of clinical experience identified latent MTrPs in the right and left UTMs by manual palpation. The diagnostic criterion for latent MTrPs was the presence of hypersensitivity tender points in the muscle taut band [3]. This procedure involved two steps: First, a palpable taut band was located via flat (cross-fiber) palpation. In this procedure, the assessor gently pushed their thumb pad inward, then slid it perpendicular to the muscle fibers. Second, the assessor systematically palpated within the taut band to identify potentially hypersensitive areas, then applied sustained and gradually increasing pressure for 3–5 seconds to confirm heightened sensitivity. The subject was asked to explicitly confirm that they experienced pain, and their response was compared to the reaction elicited by applying pressure to adjacent non-tender areas within the same muscle. We note that the inter-rater reliability when identifying MTrPs in the upper quarter muscles, including the upper trapezius muscle, has been reported to be high when assessed by experienced examiners [31].

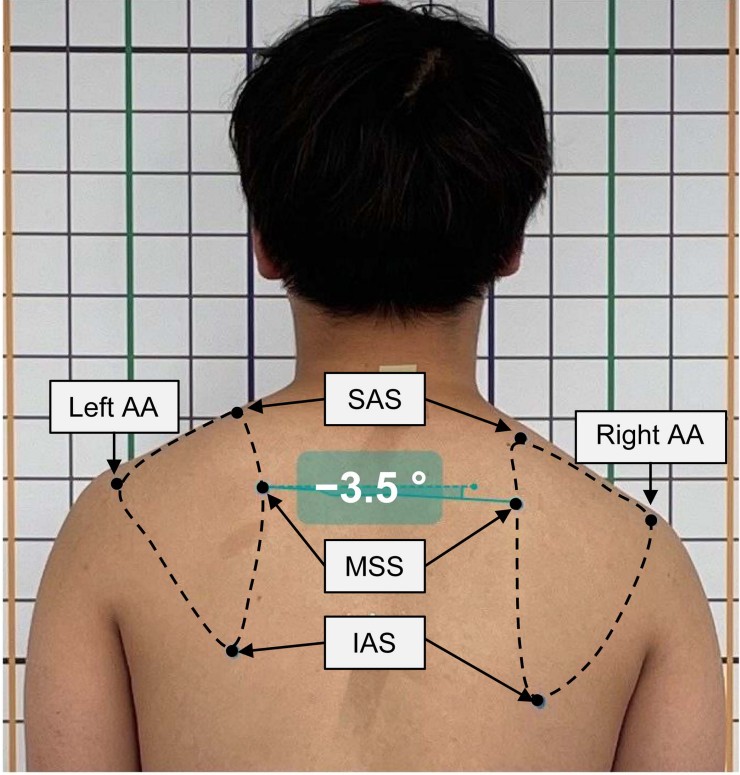

**Fig 1. Assessment of the asymmetry of the resting scapular position using a photograph.** The horizontal scapular alignment angle was defined as the angle between the horizontal line and the straight line connecting the left and right medial spine of the scapula. The positive sign of the angle indicates that the position of the right scapula is higher than that of the left scapula. Conversely, the negative sign indicates that the position of the right scapula is lower than that of the left scapula. Key anatomical landmarks of the scapula are depicted in the photograph. The dashed line indicates the (virtual) contour of the scapula. MSS: Medial spine of the scapula, AA: Acromial angle, SAS: Superior angle of scapula, IAS: Inferior angle of scapula.

## Sample size

A preliminary study with asymptomatic participants was conducted to estimate the sample size before performing the present study. The calculated effect size was 1.51 (Cohen's d) [32]. Based on the effect size, the sample size was determined using G*Power (version 3.1) [33]. With a significance level of $a = 0.05$ (two-tailed) and a statistical power of $1 − b = 0.80$, the required sample size was a total of 18 participants. Considering the withdrawal rate, the sample size should include 32 participants.

## Statistical analysis

Data are presented as mean and standard error. The normal distribution of the quantitative data were assessed using the Shapiro–Wilk test ($p > 0.05$). The participants were divided into two groups based on the presence of latent MTrPs in the UTM on each side (participants with and without latent MTrPs) to analyze whether the degree of asymmetry of the resting scapular position is related to the presence of latent MTrPs in the UTM. The difference in the participant characteristics and the HSAA between the two groups was analyzed using Student's t-test, Mann–Whitney U test, and chi-square test. In addition, to analyze the predicted factor for the degree of asymmetry of the resting scapular position, multiple regression analysis with the forced entry method was performed to predict the HSAA from the three predictor variables, namely, dominant arm, presence of latent MTrPs in the right UTM, and presence of latent MTrPs in the left UTM.

The statistical analysis was performed at a confidence level of 95%. A p-value <0.05 indicates significance. All data analyses were performed using Jamovi version 2.3.15 [34].

## Results

### Baseline characteristics

A total of 32 asymptomatic adult men were included in this study. Table 1 shows the participant characteristics, HSAA, and presence and number of latent MTrPs in the UTM. Of the 32 participants, 27 (84%) were right-arm dominant. The HSAA was −0.76±0.58. Latent MTrPs were present in 53.13% and 40.63% of the right and left UTMs, respectively.

### Influence of the presence of MTrPs in the UTM on the asymmetry of the resting scapular position

An analysis was performed to investigate whether the degree of asymmetry of the resting scapular position is related to the presence of latent MTrPs in the UTM. Table 2 shows the characteristics of the participants in each group, which were divided based on the presence or absence of latent MTrPs in the right UTM. No significant differences in participant characteristics were found between the two groups (Student's t-test, Mann–Whitney U test, and chi-square test, $P > 0.05$). Fig 2 shows a comparison of the HSAA between the two groups, which were divided based on the presence or absence of latent MTrPs in the right UTM. The HSAA of the participants with latent MTrPs (N = 17) was significantly lower than that of the participants without latent MTrPs (N = 15) in the right UTM (−0.29±0.79 vs. 0.97±0.64: Student's t-test, $p < 0.05$). Thus, the degree of depression of the right scapula relative to the left scapula was greater in participants with latent MTrPs than in those without latent MTrPs in the right UTM. The data were also analyzed based on the presence of latent MTrPs in the left UTMs, showing no significant difference in the characteristics and HSAA of the participants with (N = 13) and without (N = 19) latent MTrP (S1 Table).

**Table 1. Subject characteristics, asymmetry of the resting scapular position, and presence and number of latent MTrPs in the UTMs (n = 32).**

|  | n | % | Minimum | Maximal | Mean±SEM |
|---|---|---|---|---|---|
| Age (year) |  |  | 20 | 41 | 26.28±1.10 |
| Height (cm) |  |  | 163 | 180 | 169.94±0.70 |
| Weight (kg) |  |  | 54 | 90 | 68.90±1.51 |
| Dominant arm (R/L) | 27/5 | 84.38/15.62 |  |  |  |
| HSAA (°) |  |  | −8.03 | 5.26 | −0.76±0.58 |
| L-MTrPs in the right UTM | 17 | 53.13 | 0 | 2 | 0.66±0.13 |
| L-MTrPs in the left UTM | 13 | 40.63 | 0 | 2 | 0.53±0.13 |

HSAA, horizontal scapular alignment angle; L-MTrPs, latent myofascial trigger point; SEM, standard error of the mean; UTM, upper trapezius muscle.

**Table 2. Characteristics of the participants in the two groups, divided based on the presence of latent MTrPs in the right UTM.**

|  | No L-MTrP (n = 15) | L-MTrPs (n = 17) | p |
|---|---|---|---|
| Age (year) | 25.73±1.33 | 26.76±1.70 | >0.05[a] |
| Height (cm) | 169.73±0.92 | 170.12±1.05 | >0.05[b] |
| Weight (kg) | 68.07±2.24 | 69.65±2.04 | >0.05[b] |
| Dominant arm (R/L) | 13/2 | 14/3 | >0.05[c] |

L-MTrPs, latent myofascial trigger points.

[a]Assessed using Mann–Whitney U test.

[b]Assessed using Student's t-test.

[c]Assessed using the chi-square test.

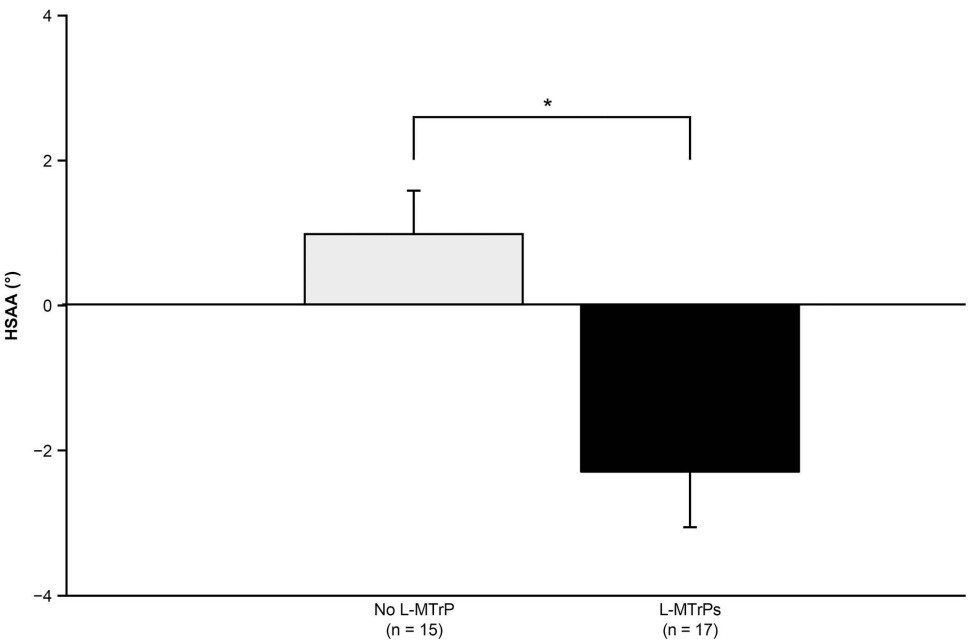

**Fig 2. Comparison of the asymmetry of the resting scapular position based on the latent myofascial trigger points (MTrPs) in the right upper trapezius muscle (UTM).** The horizontal scapular alignment angle was significantly lower in participants with latent MTrPs (L-MTrPs) in the right UTM than in those without latent MTrPs. The error bars indicate standard errors. *p < 0.05 (Student's t-test).

### Predictive factors of the asymmetry of the resting scapular position

The predictive factors of the asymmetry of the resting scapular position were analyzed using multiple regression analysis. The dominant arm was considered a factor that influenced the scapular position. The model with these variables significantly predicted the HSAA ($F(3, 28) = 5.30$, $p = 0.005$, $R^2 = 0.36$). The dominant arm and the presence of latent MTrPs in the right UTM significantly contributed to the prediction ($\beta = -2.92$, $p < 0.05$; $\beta = -2.79$, $p < 0.05$, respectively). Conversely, the presence of latent MTrPs in the left UTM did not significantly contribute to the prediction ($\beta = -1.16$, $p > 0.05$).

### Discussion

This study revealed that the presence of latent MTrPs in the right UTM and dominant arm influenced the degree of asymmetry of the resting scapular position in asymptomatic men when standing.

The results showed that the degree of depression of the right scapular relative to the left scapula was higher in participants with latent MTrPs than in those without latent MTrPs in the right UTM (Fig 2). A study reported a significant positive correlation between scapular depression and stiffness of the UTM in asymptomatic individuals [35]. Furthermore, mechanical hypersensitivity in the UTM was observed in asymptomatic individuals with a depressed scapula in the UTM [36–38]. Conversely, a study that assessed tissue stiffness using imaging techniques reported that muscle stiffness in active and latent MTrPs was higher than that in the healthy tissue [39]. Furthermore, mechanical hypersensitivity was found in the latent MTrP region [3]. These findings indicate that the increased stiffness and hypersensitivity in the UTM were common features of both scapular depression and latent MTrPs in the UTM.

A depressed scapular position results in an extended UTM position, increasing muscle tone [9,40]. The sustained localized muscle contractions generated the MTrPs [1–3]. Thus, scapular depression may overload the UTM, contributing to the formation of latent MTrPs. Conversely, latent MTrPs result in muscle weakness, accelerated muscle fatigability, changes in muscle recruitment patterns, imbalance in muscle strength and tension, and limited joint range of motion

[20,21,41]. The depression of the scapular position was suggested to be caused by the weakness of the UTM [29]. Therefore, muscle weakness caused by latent MTrPs may lead to further changes in the scapular position, creating a vicious cycle of abnormal postural alignment and development of latent MTrPs. This cycle may impose sustained mechanical stress on the muscles, leading to the transition from latent to active MTrPs and pain onset. Over time, this can result in chronic musculoskeletal pain due to the perpetuation of MTrPs.

Multiple regression analyses revealed that the dominant arm influenced the asymmetry of the resting scapular position. In right-arm dominant participants, the right scapula was more depressed than the left. Previous studies have shown that the resting scapular position of the dominant side was more depressed than that of the contralateral side in asymptomatic young adults and overhead athletes [9,30,42,43]. Dominant arms are often used in performing daily living activities and overhead throwing sports activities [44–46]. The stiffness of the UTM on the dominant side was higher than that on the nondominant side and was significantly and positively correlated with scapular depression [35]. Changes in the resting scapular position and MTrP development were attributed to sustained muscle contractions by repetitive movements [47,48]. Thus, sustained UTM contractions in the dominant arm may generate latent MTrPs, resulting in a depressed scapular resting position.

This study has some limitations. First, this study did not include asymptomatic women and people with musculoskeletal pain. Second, whole-body postural alignment, such as the spinal column and pelvis, and latent MTrPs in whole-body muscles related to postural stabilization were not assessed. Therefore, further studies are needed to examine the association between whole-body postural alignment and the presence of MTrPs in whole-muscle-related postural stabilization in symptomatic and asymptomatic individuals. Third, this study did not screen participants for physical activity level or having a history of dominant-side training. Either factor could potentially influence scapular resting position and kinematics. Therefore, future studies should consider controlling for these variables. For example, this could involve recruiting participants from specific athletic populations and/or by quantifying participant activity levels. Fourth, the causal relationship between changes in the scapular position and latent MTrPs in the UTM was not clarified. Therefore, further studies are needed to assess the effect of treatment on latent MTrPs on the scapular position to elucidate the causal relationship.

## Conclusion

This study shows that the asymmetry of the resting scapular position is significantly associated with latent MTrPs in the right UTM of asymptomatic adults. The present results and those of previous studies indicate that the relationship between the presence of latent MTrPs and body postural misalignment may contribute to the onset and persistence of musculoskeletal pain.

## Supporting information

**S1 Table. Characteristics of the participants and asymmetry of the resting scapular position of the two groups, which were divided based on the presence of latent myofascial trigger points in the left UTM.**
(DOCX)

## Acknowledgments

We would like to thank Enago (www.enago.com) for the English language editing.

## Author contributions

**Conceptualization:** Yasuhiro Aki, Kouichi Takamoto.

**Formal analysis:** Kouichi Takamoto.

**Investigation:** Yasuhiro Aki, Ippei Okino, Genki Tsuyama, Yoshiyasu Tanitsu.

**Methodology:** Kouichi Takamoto.

**Project administration:** Kouichi Takamoto.

**Supervision:** Kouichi Takamoto.

**Writing – original draft:** Yasuhiro Aki.

**Writing – review & editing:** Hisao Nishijo, Kouichi Takamoto.

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
