## [Decision Letter · Decision Letter 0]

16 Sep 2025

Dear Dr. Takamoto,

Thank you for submitting your manuscript to PLOS ONE. After careful consideration, we feel that it has merit but does not fully meet PLOS ONE’s publication criteria as it currently stands. Therefore, we invite you to submit a revised version of the manuscript that addresses the points raised during the review process.

**Diagnostic Method of Latent MTrPs**The manuscript does not clearly specify whether latent MTrPs were identified through palpation or ultrasound. Please detail the diagnostic criteria (e.g., Travell and Simons’ standards), techniques used, number of examiners, and any reliability checks. This is essential for methodological rigor.**Update and Strengthen Literature**The introduction relies heavily on older sources. To strengthen the rationale and ensure alignment with current science, please incorporate recent literature on myofascial dysfunction and postural relationships, such as:https://doi.org/10.1016/j.jbmt.2023.12.005<svg class="block h-[0.75em] w-[0.75em] stroke-current stroke-[0.75]" data-rtl-flip=“” fill="currentColor" height="20" viewbox="0 0 20 20" width="20" xmlns="http://www.w3.org/2000/svg"><path d="M14.3349 13.3301V6.60645L5.47065 15.4707C5.21095 15.7304 4.78895 15.7304 4.52925 15.4707C4.26955 15.211 4.26955 14.789 4.52925 14.5293L13.3935 5.66504H6.66011C6.29284 5.66504 5.99507 5.36727 5.99507 5C5.99507 4.63273 6.29284 4.33496 6.66011 4.33496H14.9999L15.1337 4.34863C15.4369 4.41057 15.665 4.67857 15.665 5V13.3301C15.6649 13.6973 15.3672 13.9951 14.9999 13.9951C14.6327 13.9951 14.335 13.6973 14.3349 13.3301Z"></path></svg>https://doi.org/10.1186/s13102-023-00719-y<svg class="block h-[0.75em] w-[0.75em] stroke-current stroke-[0.75]" data-rtl-flip=“” fill="currentColor" height="20" viewbox="0 0 20 20" width="20" xmlns="http://www.w3.org/2000/svg"><path d="M14.3349 13.3301V6.60645L5.47065 15.4707C5.21095 15.7304 4.78895 15.7304 4.52925 15.4707C4.26955 15.211 4.26955 14.789 4.52925 14.5293L13.3935 5.66504H6.66011C6.29284 5.66504 5.99507 5.36727 5.99507 5C5.99507 4.63273 6.29284 4.33496 6.66011 4.33496H14.9999L15.1337 4.34863C15.4369 4.41057 15.665 4.67857 15.665 5V13.3301C15.6649 13.6973 15.3672 13.9951 14.9999 13.9951C14.6327 13.9951 14.335 13.6973 14.3349 13.3301Z"></path></svg>https://doi.org/10.3390/s24030718<svg class="block h-[0.75em] w-[0.75em] stroke-current stroke-[0.75]" data-rtl-flip=“” fill="currentColor" height="20" viewbox="0 0 20 20" width="20" xmlns="http://www.w3.org/2000/svg"><path d="M14.3349 13.3301V6.60645L5.47065 15.4707C5.21095 15.7304 4.78895 15.7304 4.52925 15.4707C4.26955 15.211 4.26955 14.789 4.52925 14.5293L13.3935 5.66504H6.66011C6.29284 5.66504 5.99507 5.36727 5.99507 5C5.99507 4.63273 6.29284 4.33496 6.66011 4.33496H14.9999L15.1337 4.34863C15.4369 4.41057 15.665 4.67857 15.665 5V13.3301C15.6649 13.6973 15.3672 13.9951 14.9999 13.9951C14.6327 13.9951 14.335 13.6973 14.3349 13.3301Z"></path></svg>https://doi.org/10.3389/fspor.2024.1412412<svg class="block h-[0.75em] w-[0.75em] stroke-current stroke-[0.75]" data-rtl-flip=“” fill="currentColor" height="20" viewbox="0 0 20 20" width="20" xmlns="http://www.w3.org/2000/svg"><path d="M14.3349 13.3301V6.60645L5.47065 15.4707C5.21095 15.7304 4.78895 15.7304 4.52925 15.4707C4.26955 15.211 4.26955 14.789 4.52925 14.5293L13.3935 5.66504H6.66011C6.29284 5.66504 5.99507 5.36727 5.99507 5C5.99507 4.63273 6.29284 4.33496 6.66011 4.33496H14.9999L15.1337 4.34863C15.4369 4.41057 15.665 4.67857 15.665 5V13.3301C15.6649 13.6973 15.3672 13.9951 14.9999 13.9951C14.6327 13.9951 14.335 13.6973 14.3349 13.3301Z"></path></svg>

**Discussion Formatting**Subheadings in the discussion section disrupt the flow. Please remove them and present the discussion as a cohesive narrative in line with PLOS ONE’s style.Revise **Line 47** for grammatical accuracy: “Simons explained…” rather than “They explained…”.In **Figure 2 caption** , replace “patients” with “participants” to reflect the healthy cohort.Clarify in **Methods/Ethics** whether participants were screened for physical activity levels or dominant-side training, as these factors may influence scapular position.Conduct a thorough **language and style edit** , ideally by a native English speaker with biomedical expertise, to improve clarity and polish.

Reviewer 2: 

To enhance the educational impact, I recommend including a figure or diagram that transparently overlays the anatomical reference points of the scapula on the photo provided. This would help readers and clinicians better visualize the relationship between scapular alignment and trigger point localization

We look forward to receiving your revised manuscript.

Kind regards,

Prateek Srivastav

Academic Editor

PLOS ONE

Journal Requirements:

“This work was supported by the Givers Corporation (https://givers.co.jp/) and a Grant-in-Aid for Scientific Research (C) (23K10467) award to K.T. (https://www.jsps.go.jp/j-grantsinaid/). Y.A., I.O., G.T., and Y.T. are employees of Givers Corporation.”

3. We note that you have indicated that there are restrictions to data sharing for this study. PLOS only allows data to be available upon request if there are legal or ethical restrictions on sharing data publicly. For more information on unacceptable data access restrictions, please see http://journals.plos.org/plosone/s/data-availability#loc-unacceptable-data-access-restrictions .   

Reviewers' comments:

Reviewer's Responses to Questions

**Comments to the Author**

1. Is the manuscript technically sound, and do the data support the conclusions?

Reviewer #1: Yes

Reviewer #2: Yes

2. Has the statistical analysis been performed appropriately and rigorously?

Reviewer #1: Yes

Reviewer #2: Yes

3. Have the authors made all data underlying the findings in their manuscript fully available?

Reviewer #1: Yes

Reviewer #2: Yes

4. Is the manuscript presented in an intelligible fashion and written in standard English?

Reviewer #1: Yes

Reviewer #2: Yes

Reviewer #1: Review of the manuscript titled: Relationship between the asymmetry of the resting scapular position and the prevalence of latent myofascial trigger points in the trapezius muscle in asymptomatic adults. The topic is of importance in musculoskeletal research and clinical biomechanics, particularly given the increasing interest in subclinical neuromuscular imbalances and their potential role in the development of pain syndromes. The manuscript is well-structured and methodologically sound. The sample size calculation is transparently reported and justified, and the statistical analysis appears appropriate for the research question. The manuscript’s findings, particularly the association between scapular depression and latent MTrPs on the dominant side, provide important insights into the pathophysiological continuum between muscle dysfunction and postural misalignment in asymptomatic individuals. The conclusion is cautious yet meaningful, suggesting a potential link that merits further investigation in longitudinal or interventional designs.

My comments:

- The manuscript does not clearly specify whether the diagnosis of latent MTrPs was conducted via palpation or USG. Although it is mentioned that an experienced clinician performed the assessment, the methodological rigour would benefit from a clearer explanation of the diagnostic criteria and techniques used (e.g., application of Travell and Simons’ criteria, number of examiners, intra-rater reliability). This should be elaborated in the Methods section.

- While the theoretical background is generally well presented, the authors rely heavily on outdated sources (e.g., Simons et al., 1998). The mechanistic model of MTrP development, as outlined in the introduction, could be more robustly contextualised using recent literature. I encourage the authors to incorporate findings from the following recent works, which offer updated insights into myofascial dysfunction, postural relationships, and muscle stiffness: https://doi.org/10.1016/j.jbmt.2023.12.005 ; https://doi.org/10.1186/s13102-023-00719-y ; https://doi.org/10.3390/s24030718 or https://doi.org/10.3389/fspor.2024.1412412. Incorporating these will strengthen the rationale and ensure the discussion reflects the current state of the science.

-The discussion is well-written and clinically insightful. However, the current use of subheadings within the discussion section feels unnecessary and somewhat interrupts the narrative flow. I recommend removing these subheadings and integrating the content into a continuous, cohesive discussion as is standard in PLOS ONE formatting.

- Line 47: The sentence “Simons [1] proposed the mechanism of MTrP formation. They explained that…” should be revised for grammatical consistency. Since the cited work refers to a single author (Janet G. Travell or David G. Simons), using “he/she explained” or simply “Simons explained…” would be more appropriate.

- Figure 2 Caption: The text refers to “patients” rather than “participants” in an otherwise healthy cohort. This may cause confusion and should be corrected for terminological precision.

- Ethical Clarity: While the ethics approval is described adequately, it may be beneficial to clarify whether the participants were screened for physical activity levels or dominant-side training, given their potential influence on scapular positioning and muscle function.

- Language and Style: The manuscript is generally clear, but a thorough proofreading by a native English speaker (ideally one familiar with biomedical terminology) would further enhance readability and polish.

In conclusion, with minor revisions, particularly in clarifying methods, updating literature, and improving textual fluency, the manuscript will meet the high standards expected for publication in PLOS ONE.

Reviewer #2: This work is significant because of its practical relevance in understanding myofascial pain in the trapezius muscle region.

Scapular misalignment can create stress in the trapezius muscle and the creation of myofascial trigger points, resulting in pain in the scapula region. Healthcare professionals can utilize this information to guide patients through clinical examinations, assisting them in treating and preventing pain in the region.

Suggestion:

You could draw a diagram of transparency in the photo you used, using the anatomical reference points of the scapula for educational purposes

**Do you want your identity to be public for this peer review?** For information about this choice, including consent withdrawal, please see our Privacy Policy

Reviewer #1: No

Reviewer #2: No

---

## [Author Response · Author response to Decision Letter 1]

7 Oct 2025

Reviewer #1, Comment 1:

Diagnostic Method of Latent MTrPs

The manuscript does not clearly specify whether latent MTrPs were identified through palpation or ultrasound. Please detail the diagnostic criteria (e.g., Travell and Simons’ standards), techniques used, number of examiners, and any reliability checks. This is essential for methodological rigor.

Response:

To address this comment, we rewrote this section and added new information to the Methods and References sections, as stated below. Detailed information regarding the diagnostic criteria used here is also provided in descriptions of other techniques used to identify latent MTrPs.

A single assessor with 10 years of clinical experience identified latent MTrPs in the right and left UTMs by manual palpation. The diagnostic criterion for latent MTrPs was the presence of hypersensitivity tender points in the muscle taut band [3]. This procedure involved two steps: First, a palpable taut band was located via flat (cross-fiber) palpation. In this procedure, the assessor gently pushed their thumb pad inward, then slid it perpendicular to the muscle fibers. Second, the assessor systematically palpated within the taut band to identify potentially hypersensitive areas, then applied sustained and gradually increasing pressure for 3–5 seconds to confirm heightened sensitivity. The subject was asked to explicitly confirm that they experienced pain, and their response was compared to the reaction elicited by applying pressure to adjacent non-tender areas within the same muscle. We note that the inter-rater reliability when identifying MTrPs in the upper quarter muscles, including the upper trapezius muscle, has been reported to be high when assessed by experienced examiners [31].

31. del Moral OM, Lacomba MT, Russell IJ, Méndez ÓS, Sánchez BS. Validity and reliability of clinical examination in the diagnosis of myofascial pain syndrome and myofascial trigger points in upper quarter muscles. Pain Medicine. 2018;19:2039-2050. doi: 10.1093/pm/pnx315.

Reviewer #1, Comment 2:

Update and Strengthen Literature

The introduction relies heavily on older sources. To strengthen the rationale and ensure alignment with current science, please incorporate recent literature on myofascial dysfunction and postural relationships, such as:

https://doi.org/10.1016/j.jbmt.2023.12.005

https://doi.org/10.1186/s13102-023-00719-y

https://doi.org/10.3390/s24030718

https://doi.org/10.3389/fspor.2024.1412412

Response:

In response to this comment, we have rewritten the introduction and cited more recent literature, as outlined below.

The mechanism of MTrP formation has been proposed to involve mechanical stress from repetitive or sustained movements, which overload the muscle and result in sustained, localized muscle contractions (muscle taut bands).

1. Lam C, Francio VT, Gustafson K, Carroll M, York A, Chadwick AL. Myofascial pain - A major player in musculoskeletal pain. Best practice & research. Best Pract Res Clin Rheumatol. 2024;38:101944. doi: 10.1016/j.berh.2024.101944.

2. Shah JP, Thaker N, Heimur J, Aredo JV, Sikdar S, Gerber, L. Myofascial trigger points then and now: A historical and scientific perspective. PM R. 2015;7:746-761. doi: 10.1016/j.pmrj.2015.01.024.

7. Edwards J. The importance of postural habits in perpetuating myofascial trigger point pain. Acupunct Med. 2005;23:77-82. doi: 10.1136/aim.23.2.77.

8. Steen JP, Jaiswal KS, Kumbhare D. Myofascial pain syndrome: An update on clinical characteristics, etiopathogenesis, diagnosis, and treatment. Muscle Nerve. 2025;71:889-910. doi: 10.1002/mus.28377.

Reviewer #1, Comment 3:

Discussion Formatting

Subheadings in the discussion section disrupt the flow. Please remove them and present the discussion as a cohesive narrative in line with PLOS ONE’s style.

Response:

As per the reviewer recommendation, we have removed all subheadings in the Discussion section.

Reviewer #1, Comment 4:

Revise Line 47 for grammatical accuracy: “Simons explained…” rather than “They explained…”.

Response:

Based on other comments regarding of nature of the scientific literature, this line has been rewritten as follows:

The mechanism of MTrP formation has been proposed to involve mechanical stress from repetitive or sustained movements, which overload the muscle and result in sustained, localized muscle contractions (muscle taut bands) [1-3].

Reviewer #1, Comment 5:

In Figure 2 caption, replace “patients” with “participants” to reflect the healthy cohort.

Response:

In response to this comment, we have corrected “patients” to “participants”.

Reviewer #1, Comment 6:

Clarify in Methods/Ethics whether participants were screened for physical activity levels or dominant-side training, as these factors may influence scapular position.

Response:

According to the comment, we add the information to the methods as stated below:

No specific exclusion criteria were applied based on physical activity levels or dominant-side sport-specific training.

Additionally, we add the information to the discussion as stated below:

Third, this study did not screen participants for physical activity level or having a history of dominant-side training. Either factor could potentially influence scapular resting position and kinematics. Therefore, future studies should consider controlling for these variables. For example, this could involve recruiting participants from specific athletic populations and/or by quantifying participant activity levels.

Reviewer #1, Comment 7:

Conduct a thorough language and style edit, ideally by a native English speaker with biomedical expertise, to improve clarity and polish.

Response:

Thank you for your valuable suggestion. We have conducted a thorough language and style revision of the entire manuscript with the assistance of a native English speaker with expertise in biomedical writing. We have carefully edited the text to improve clarity, grammar, and overall readability. All changes have been incorporated in the revised version of the manuscript.

Reviewer #2, Comment 1:

To enhance the educational impact, I recommend including a figure or diagram that transparently overlays the anatomical reference points of the scapula on the photo provided. This would help readers and clinicians better visualize the relationship between scapular alignment and trigger point localization.

Response:

As suggested in this comment, we have included a revised Figure in the latest version of our manuscript. We have also rewritten the Figure Legend as stated below:

Key anatomical landmarks of the scapula are depicted in the photograph. The dashed line indicates the (virtual) contour of the scapula. MSS: Medial spine of the scapula, AA: Acromial angle, SAS: Superior angle of scapula, IAS: Inferior angle of scapula.

---

## [Editor Report · Decision Letter 1]

8 Oct 2025

Relationship between the asymmetry of the resting scapular position and the prevalence of latent myofascial trigger points in the trapezius muscle in asymptomatic 

PONE-D-25-37154R1

Dear Dr. Takamoto,

We’re pleased to inform you that your manuscript has been judged scientifically suitable for publication and will be formally accepted for publication once it meets all outstanding technical requirements.

Kind regards,

Prateek Srivastav

Academic Editor

PLOS ONE
---

## [Editor Report · Acceptance letter]

PONE-D-25-37154R1

PLOS ONE

Dear Dr. Takamoto,

I'm pleased to inform you that your manuscript has been deemed suitable for publication in PLOS ONE. Congratulations! Your manuscript is now being handed over to our production team.

Kind regards,

on behalf of

Dr. Prateek Srivastav

Academic Editor

PLOS ONE